# Benchmarking Compositional generalisation for Learning Inter-atomic Potentials

## Abstract

Inter-atomic potentials play an important role for modelling molecular dynamics. Unfortunately, traditional methods for computing such potentials are computationally heavy. In recent years, the idea of using neural networks to approximate these computations has gained in popularity, and a variety of Graph Neural Networks and Transformer based methods have been proposed for this purpose. Recent approaches provide highly accurate estimates, but they are typically trained and tested on the same molecules. It thus remains unclear whether these models mostly learn to interpolate the training labels, or whether their physically-informed designs actually allow them to capture the underlying principles. To address this gap, we propose a benchmark consisting of four tasks that each require some form of compositional generalisation. Training and testing involves separate molecules, but the training data is chosen such that generalisation to the test examples should be feasible for models that learn the physical principles. Our empirical analysis shows that the considered tasks are highly challenging for state-of-the-art models, with errors for out-of-distribution examples often being orders of magnitude higher than for in-distribution examples.

## 1 Introduction

Inter-atomic potentials and their associated force fields are central to Molecular Dynamics (MD) simulation, a widely used technique that provides atomistic insights into physical phenomena, enabling applications such as drug design and material discovery (Alder & Wainwright, 1959; Schlick, 2010; Rahman, 1964). Inter-atomic potentials, often parameterised using Density Functional Theory (DFT) for high accuracy despite its computational cost, approximate the potential energy surface (PES) and enable long-timescale MD of materials and biological systems (Frenkel & Smit, 2023). In recent years, the use of machine learning models for predicting force fields has become increasingly widespread, with Graph Neural Networks (GNNs) and Transformer based architectures being particularly popular (Gilmer et al., 2017; Liao et al., 2023). Such Machine Learning Force Fields (MLFFs) can be computed efficiently, and the predictions of recent state-of-the-art models are accurate enough for many applications (Batatia et al., 2022). However, they are not entirely independent of the computationally expensive methods they seek to replace. MLFFs still rely on high-accuracy reference calculations for their training data, which typically consists of optimisation trajectories and initial MD segments. The trained MLFFs are then used to perform more or less stable, long-timescale MD simulations (Neumann et al., 2024). While the architecture of MLFFs is often physically-informed, it is unclear to what extent they actually learn the underlying physical principles, or whether they essentially learn to interpolate between the training examples. This question matters, because models that learn the underlying principles would be expected to generalise better. Ideally, we want models that generalise beyond the molecules that they have been trained on, but this is something that is not evaluated in standard benchmarks for MLFFs.

In this paper, we introduce the *Generalisation for Molecular Dynamics* (GMD-25) benchmark. GMD-25 consists of four tasks, which test for different aspects of compositional generalisation (Hupkes et al., 2020), as illustrated in Figure 1. Each task consists of a training and an *out-of-distribution* (OOD) test set consisting of MD trajectories. Crucially, in contrast to standard practice, the trajectories in the training and test set come from different molecules. The training molecules are chosen such that MLFFs should, in principle, be able to generalise to the test molecules. For instance, for the Length Extrapolation task, we train models on MD trajectories of linear alkanes

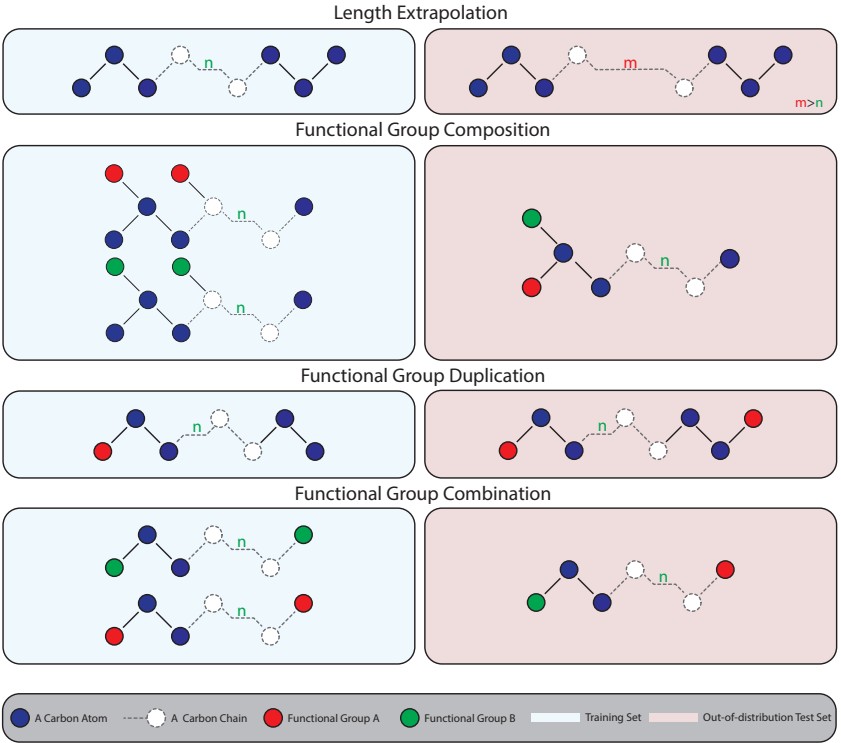

Figure 1: An overview over the generalisation tasks covered by our GMD-25 benchmark. In each task, the training set consists of a number of molecules, differing in the length of the carbon chain, and in some cases the functional group(s). The out-of-distribution test data includes similar molecules, but with a longer carbon chain and/or a different combination of functional groups. In each case, the training data covers all the basic components that the model needs to interpret the test molecules.

with $\{2, ..., 6\}$ carbon atoms and subsequently test on longer, unseen alkanes with $\{7, ..., 13\}$ carbon atoms. The benchmark's ab initio molecular dynamics (AIMD) trajectories were generated using a novel toolkit, also introduced here, which is designed to make the benchmark easily extensible.

We evaluate several popular MLFFs on our benchmark, including SchNet (Schütt et al., 2018), PAINN (Schütt et al., 2021), DimeNet$^{++}$ (Gasteiger et al., 2020), GemNet (Gasteiger et al., 2022; 2021), and Equivariant Transformer V2 (EquiFormerV2) (Liao et al., 2023). These models represent a diverse range of design choices in molecular representation learning, encompassing invariant and equivariant models, and both GNN and Transformer-based approaches. Our analysis shows that the considered generalisation tasks are highly challenging for these models, with errors on the test molecules sometimes being several orders of magnitude higher than errors on in-distribution (ID) examples (i.e. previously unseen configurations of the training molecules). Importantly, the models that perform best on ID examples are not always the models that generalise best to OOD examples, suggesting that the most popular approaches may not always learn the most physically plausible models. With the proposed benchmark, we hope to encourage a shift towards the development of models with better generalisability.

## 2 RELATED WORK

### 2.1 MACHINE LEARNING FORCE FIELDS

The study of MLFFs dates back to Blank et al. (1995). While early models relied on manually engineered functions to represent local atomic environments (Behler & Parrinello, 2007; Zhang et al., 2018; 2019), more recent approaches have used learnable descriptors which are inferred directly from the data, typically with the use of message-passing neural networks (MPNNs) (Thölke & De Fabritiis,

2022; Batatia et al., 2022; Schütt et al., 2018). Within this category, a key distinction arises based on how geometric symmetries are handled. A conceptually straightforward approach is to train models on invariant features only. For instance, models such as SchNet (Schütt et al., 2018) operate on pairwise radial distances, which are invariant to rotations and translations, to predict energy potentials. Force vectors can then be obtained by computing the gradient of the network. A more powerful approach involves learning SE(3)-equivariant features (Unke et al., 2021), which transform predictably with the system's coordinates, making it possible, for instance, to directly predict forces. One prominent line of work, encompassing Tensor Field Networks (TFNs) and NequIP (Batzner et al., 2022; Fuchs et al., 2020; Thomas et al., 2018), achieves equivariance through mathematically rigorous formulations based on spherical harmonics and irreducible representations. Models such as EGNN (Satorras et al., 2021) and TorchMDNet (Thölke & De Fabritiis, 2022) offer a more efficient alternative, by directly operating on vectors in 3D space. More recently, the state-of-the-art has been advanced by Transformer-based architectures such as EquiFormerV2 (Liao et al., 2023), which combine equivariant principles with attention to achieve new levels of accuracy and scalability, defining the current frontier for MLFFs.

## 2.2 COMPOSITIONAL GENERALISATION

There is a long-standing debate about whether neural networks are capable of compositional generalisation (Fodor & Pylyshyn, 1988), i.e. whether they are capable of solving problems that require combining solutions to sub-problems in novel ways. The idea of algorithmic alignment seems to play an important role, where neural networks are found to generalise well if their architecture is sufficiently similar to the structure of the algorithm they are supposed to learn (Xu et al., 2020; Zhou et al., 2024; Khalid & Schockaert, 2025). This view suggests that using carefully designed physics-informed architectures might be important for MLFFs to generalise well.

Within the context of molecular machine learning, it has been observed that GNNs trained on specific molecular datasets often struggle to predict properties for molecules with different structural characteristics or scaffolds (Ektefaie et al., 2024). This problem is of particular importance in drug discovery, where identifying novel compounds outside the known chemical space is essential (Li et al., 2022). In response, benchmarks such as DrugOOD (Ji et al., 2023) have been introduced to assess the robustness of GNNs in drug–target binding prediction tasks, by employing scaffold- and protein-family based train-test splits. Similarly, in materials science, MatBench (Omee et al., 2024) adapted standard datasets, creating extrapolative training-test splits based on compositional and structural groups. Their analysis found GNNs to struggle with predicting the properties of previously unseen materials. The BOOM benchmark (Antoniuk et al., 2025) studies generalisation from a different angle. Unlike the compositional and structural tasks that are central to our study, BOOM offers a property-centric perspective on extrapolation. Their challenge is framed not in terms of unseen molecular structures, but rather in terms of unseen property values. The authors construct their test sets using molecules from the tail ends of the target property's distribution, explicitly assessing whether a model can generalise beyond the numerical range of its training data.

## 2.3 BENCHMARKING MLFFS

Molecular dynamics (MD) datasets have played a central role in benchmarking MLFFs. The MD17 dataset (Chmiela et al., 2017) consists of *ab initio* MD trajectories for a small set of molecular systems at equilibrium, with configurations sampled near their minima. While MD17 has driven early model development, its limited diversity and coverage has prompted the introduction of more comprehensive datasets. For example, WS22 (Pinheiro Jr et al., 2023) expands configurational diversity using Wigner sampling and interpolation for small organic molecules. Transition1x (Schreiner et al., 2022) extends the setting further, comprising approximately 9.6M energy and force labels over ~10k reaction pathways, and demonstrates that models trained solely on equilibrium data, such as MD17, QM9 (Ramakrishnan et al., 2014) and ANI-1 (Smith et al., 2017), fail to generalise to transition-state geometries. To probe photochemical processes, xxMD (Pengmei et al., 2024) provides excited-state reactive trajectories, where standard MLFFs exhibit significantly higher errors than on ground-state datasets. At the other end of the spectrum, MD22 (Chmiela et al., 2023) introduces MD datasets for large, flexible systems (42–370 atoms), including peptides and carbon nanotubes, enabling evaluation on highly nonlocal molecular dynamics. These existing benchmarks were aimed at increasing the number of training examples and widening the range of molecules. In contrast, in our

GMD-25 benchmark, chemical subspaces are selected to systematically assess models' compositional generalisation capabilities, allowing for smaller and more focused training sets.

## 3 THE GMD-25 BENCHMARK

We introduce **GMD-25**, a systematic and extensible benchmark designed to evaluate the generalisation capabilities of MLFFs. Unlike previous datasets that either narrowly focus on equilibrium dynamics (e.g., MD17) or aim for broad chemical coverage without controlled evaluation tasks (e.g., ANI-1), GMD-25 is constructed to facilitate systematic analysis of compositional generalisation. The dataset consists of examples from the ten molecular groups based on substituted linear (alkyl) carbon chains, extended via different functional group(s). Each group contains 5-16 chemically related molecules, chosen to probe specific generalisation challenges. For each molecule, we provide two AIMD trajectories. The energy and forces were calculated using the GNF2-xTB semi-empirical tight-binding approach (Bannwarth et al., 2019), a method which is known for its balance between computational efficiency and accuracy, yielding robust labels for modeling organic molecules. More details on the dataset can be found in the appendix.

### 3.1 EVALUATION TASKS

The evaluation tasks in GMD-25 focus on two central aspects of compositional generalisation, namely *length generalisation* and *systematicity* (Hupkes et al., 2020). Length generalisation refers to the ability of a model that is trained on sequence data to generalise to longer sequences than the ones seen during training. Length generalisation plays an important role in Natural Language Processing, for instance, as models should ideally generalise to sentences of arbitrary length. In the case of molecules, we evaluate length generalisation with respect to the length of carbon chains in linear alkanes. Systematicity refers to the ability of a model to combine sub-components of problem instances in novel ways. We evaluate this ability by varying the combination of functional groups in molecules, testing molecules on previously unseen combinations, while ensuring that each of the individual functional groups appears in the training data. For all tasks, the aim is to learn a model that can predict potential energy and forces.

**Task 1: Length extrapolation** In this task, models are evaluated on their ability to extrapolate to larger molecules than those they were trained on. We consider two variants, called *base* and *augmented*. For the base variant, alkanes are used in both the training and the test set. The training set consists of trajectories for alkanes with carbon chain lengths in $\{2, ..., 6\}$ (one trajectory per molecule). Each of these five trajectories contains around 2000 snapshots. The out-of-distribution test set consists of trajectories of alkanes with carbon chain lengths in $\{7, ..., 13\}$ (with again 2000 snapshots per trajectory). To analyse the performance of the trained models on in-distribution samples, we also created an in-distribution test set, containing unseen snapshots of alkanes with carbon chain lengths in $\{2, ..., 6\}$, taken from different trajectories than those used from the training set.

In the augmented variant, the training data contains *alcohols* with carbon chain lengths in $\{2, 3\} \cup \{9, ..., 15\}$ and *carboxylic acids* with carbon chain lengths in $\{4, ..., 8\}$. The out-of-distribution test set contains *alcohols* with carbon chain length in $\{4, ..., 8\}$ and *carboxylic acids* with lengths in $\{2, 3\} \cup \{9, ..., 15\}$. In other words, the model is tested on *combinations* of lengths and functional groups that it has not seen before. However, since the model has seen all carbon chain lengths during training, we might expect this augmented variant to be easier than the base variant.

The ability to extrapolate to larger molecules is critical for the practical application of MLFFs. For instance, in drug discovery, models trained on small molecular fragments or peptides must be able to generalise to larger, more complex lead compounds (Erlanson et al., 2016) or polypeptide chains (Muller et al., 2018). The Length Extrapolation task therefore serves as a fundamental probe of whether an MLFF has learned a physically plausible representation of how inter-atomic interactions scale with distance, or if it has simply overfitted to the size distribution of the training set.

**Task 2: Functional group composition** This task assesses a model's ability to generalise to a novel functional group that is a composite of familiar moieties. We again consider a *base* and *augmented* variant of this task. For the base variant, the training set consists of *alcohol* and *aldehyde* molecules, while the test set consists of *carboxylic acid* molecules. Note that the functional group of the latter

can be seen as a composition of the functional group of the former two. In addition, we expand the training set with *complex carbonyls* and *alcohol* molecules to provide additional coverage of the special bonds from the functional groups that appear in the aforementioned molecules. For this task, we focus on molecules with carbon chain lengths in $\{4, ..., 10\}$, both for the training and for the out-of-distribution test set. For the *complex carbonyls* and *alcohol* molecules (in the training set), we also included molecules with a carbon chain length of 11. Each trajectory contains around 2000 snapshots. We again used secondary trajectories to construct an in-distribution test set.

For the augmented variant, the training data additionally contains examples of *amines* and *amides* molecules. The functional group of the latter is a composition of the functional groups of aldehyde and *amines*, hence the training data for the augmented variant includes a demonstration of how functional groups can be composed. As before, the training data also contains *alcohol*, *aldehyde*, *complex carbonyls*, and *complex alcohol* molecules, while the out-of-distribution test set contains *carboxylic acid* molecules. We might expect the augmented variant to be easier than the base variant.

The combinatorial vastness of chemical space makes it impossible to exhaustively cover all molecules in the training data. A primary goal for MLFFs is therefore to develop models that can generalise to novel molecules by learning the contributions of their constituent components. The Functional Group Composition task provides a direct and challenging evaluation of this capability. It is important to clarify that we do not expect the model to learn the chemical reaction pathway, but rather to infer the properties of the composite group from the learned effects of its constituent parts. Success on such a task would represent a significant step towards MLFFs that can truly accelerate molecular discovery.

**Task 3: Functional group duplication**   This task evaluates a model's ability to generalise from a single occurrence of a chemical motif to two occurrences of that same motif within an otherwise identical molecule. The training data contains various *monocarboxylic acids* trajectories (i.e. with one occurrence of the functional group) with carbon chain length in $\{5, ..., 10\}$, while the out-of-distribution test set contains the corresponding *dicarboxylic acids* (i.e. with two occurrence of the functional group) with identical carbon chain lengths. Each trajectory again consists of 2000 snapshots and secondary trajectories were used to create an in-distribution test set.

The ability to generalise from a single chemical motif to its repetition is crucial for many applications, from drug discovery to polymer science. For example, the therapeutic effect of a drug can depend on its interaction with systems containing repeating biological units (Mammen et al., 1998), while the properties of polymers are defined by the repetition of a single monomer—a generalisation so challenging that leading methods often rely on techniques like transfer learning to bridge the knowledge gap from small molecules (St John et al., 2019). However, this reliance on pre-training is a critical limitation, as a large, relevant dataset is not always available for novel chemical spaces. Furthermore, it deviates from the primary goal of MLFFs: to create a universally applicable potential, analogous to general approximation methods like DFT. Predicting the properties of these larger, periodic systems is a non-trivial generalisation challenge, as interactions between repeated moieties can introduce complex, non-linear effects. The Functional Group Duplication task serves as a fundamental test of this capability. An MLFFs that can successfully solve this task would represent a significant step towards models that can accurately predict the properties of oligomers and other periodic systems, a currently challenging frontier for the field.

**Task 4: Functional group combination**   This task evaluates a model's ability to generalise to asymmetrically functionalised molecules, when being trained exclusively on symmetrically functionalised analogues. In other words, the aim is to determine if the model can learn the independent identities of different functional groups and recombine them in a novel, asymmetric configuration on a familiar scaffold. The training set contains two types of molecules: molecules with two *carboxylic acids* functional groups and molecules with two *amines*. Each group contains 8 trajectories, with carbon chain length in $\{2, ..., 9\}$, each comprising around 2000 snapshots. The out-of-distribution test set contains molecules of the same lengths that contain one *carboxylic acids* functional group and one amine. We again used secondary trajectories for in-distribution test set. By keeping the underlying molecular scaffold consistent across training and testing, this task isolates the challenge to the symbolic recombination of learned functional patterns, probing the model's capacity to handle hetero-functionalisation.

## 3.2 Toolkit implementation and workflow

Our toolkit was implemented in Python, leveraging a combination of specialised libraries with the Atomic Simulation Environment (ASE) (Larsen et al., 2017) as the core framework. Key libraries include RDKit[1] for initial structure generation, FlashMD (Bigi et al., 2025) for efficient trajectory simulations, and XTB-Python for semi-empirical calculations. The toolkit follows four steps to generate each trajectory:

**Initial Structure Generation:** The workflow begins with a molecular representation (e.g., a SMILES string), from which the RDKit package generates an initial 3D geometry.

**Initial Trajectory Generation:** Using this 3D structure, FlashMD performs a fast molecular dynamics simulation. This method provides longer simulations with less computational cost, and consequently samples a wider range of off-equilibrium configurations for each molecule. Molecules were simulated in vacuum using a Langevin thermostat (300 K) and a 16 femtosecond timestep for 200k steps. While inserting noise to coordinates and computing forces accordingly is a recently used approach (Feng et al., 2023), we believe that generating dynamic trajectories is a more consistent and transferable approach for our benchmark.

**High-Fidelity Recalculation:** Each snapshot from the initial trajectory is then refined, by recalculating the energy and forces for every frame using the more accurate GFN2-xTB method.

**Orchestration and Output:** The entire process is managed by ASE, which outputs the final dataset. The data, containing refined coordinates, forces, and total energies for each frame, is stored in a standardized format to ensure seamless integration with existing MLFFs pipelines.

To facilitate reproducibility and ease of use, the dataset is released with curated data splits and pre-processing scripts, which are provided within a companion framework forked from the first version of the `fairchem`[2] and designed for training MLFFs.[3] In total, the dataset comprises 118 molecules and 296,534 labelled geometries.

## 4 Evaluation

### 4.1 Models

We use our benchmark to evaluate a diverse set of state-of-the-art MLFFs, representing distinct architectural families. First, as an established baseline, we included SchNet (Schütt et al., 2018), a well-known invariant GNN-based model. To represent more recent advances in equivariant message passing, we selected PAINN (Schütt et al., 2021). We also evaluated models that explicitly incorporate geometric features such as dihedral angles, namely GemNet (Gasteiger et al., 2022; 2021) and DimeNet$^{++}$ (Gasteiger et al., 2020). Finally, to represent the current frontier of equivariant architectures, we included the transformer-based EquiFormerV2 (Liao et al., 2023). Note that our benchmark provides a controlled setting for evaluating the generalisation abilities of different neural network architectures. As such, we did not include any foundation models (Batatia et al., 2023) in our analysis. The latter have been pre-trained on large and diverse sets of molecules, making it harder to untangle memorisation and generalisation effects.

### 4.2 Experimental set-up

For the experimental protocol, we adopted a two-stage hyperparameter tuning strategy. Initially, models were trained using the curated default hyperparameters provided in the `fairchem` repository to establish a baseline performance. Subsequently, we conducted a Bayesian hyperparameter optimisation to ensure that each model achieved its best possible performance on the in-distribution data within our computational allowance, and to analyse the sensitivity of the models to hyperparameter selection. The resulting optimised hyperparameters can be found in the appendix.

To quantify model performance, our evaluation focuses on two primary metrics across all four tasks: the *Mean Absolute Error (MAE) on forces*, reported in eV/Å, and the *MAE on energy*, reported in

---

[1]RDKit: Open-source cheminformatics. https://www.rdkit.org/

[2]https://github.com/facebookresearch/fairchem

[3]The full toolkit will be made available upon acceptance.

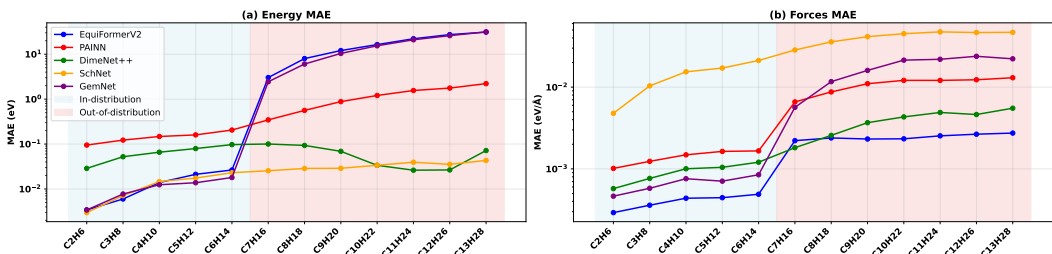

Figure 2: Results for the base variant of Length Extrapolation (Task 1). Models were trained on MD trajectories of linear alkanes with 2-6 carbon atoms (in-distribution, light blue region) and evaluated on longer, unseen chains with 7-13 carbon atoms (out-of-distribution, light red region). The figure shows: (a) MAE on total energy and (b) MAE on atomic forces. Both error metrics, displayed on a logarithmic scale, demonstrate a consistent trend across all five models, where performance degrades sharply at the distribution shift.

eV. The force MAE is calculated over all Cartesian components of atomic forces, while the energy MAE is computed on the total energy of each molecular configuration:

$$\text{MAE}_{\text{force}} = \frac{1}{3N} \sum_{i=1}^{N} \sum_{c \in \{x,y,z\}} |\hat{F}_{i,c} - F_{i,c}|, \quad \text{MAE}_{\text{energy}} = \frac{1}{M} \sum_{j=1}^{M} |\hat{E}_j - E_j|$$

where $M$ is the number of molecules in the test set and $N$ is the number of atoms across all molecules. We write $\hat{E}_j$ for the predicted energy for molecule $j$ and $E_j$ for the ground truth. Similarly, $(\hat{F}_{i,x}, \hat{F}_{i,y}, \hat{F}_{i,z})$ denotes the predicted force vector for atom $i$ and $(F_{i,x}, F_{i,y}, F_{i,z})$ is the corresponding ground truth vector. These metrics provide a comprehensive assessment of model accuracy for the fundamental quantities required in molecular dynamics simulations. Additional force analysis metrics are presented in the appendix for completeness.

### 4.3 RESULTS

**Length extrapolation** The results for the base variant of the Length Extrapolation task are presented in Figure 2. A clear trend is observed across all models: predictive accuracy deteriorates significantly at the distribution shift. *EquiFormerV2* consistently exhibits the lowest Forces MAE (panel b) However, its Energy MAE increases dramatically in the OOD region, eventually becoming the worst-performing model (panel a). Conversely, *SchNet* and *DimeNet*$^{++}$ exhibit more stable energy predictions in the OOD region, despite their weaker performance on forces. Additional force analysis metrics are provided in the appendix.

The results for the augmented variant are presented in Figure 3. For Energy MAE (panels a and c), *DimeNet*$^{++}$ and *SchNet* generalise effectively, maintaining stable and low error across both ID and OOD regions. In contrast, *EquiFormerV2* fails on this metric, with its error increasing by an order of magnitude in the OOD region. However, for Forces MAE (Figure 3 panels b and d), *EquiFormerV2* was the top performer, exhibiting a generally small generalisation gap, while *SchNet* and *GemNet* performed poorly (both ID and OOD).

**Functional group composition** As shown in Figure 4 (panels a–d), all models fail to generalise for this task, both in the base and the augmented variants. When looking at forces MAE in the base variant (panel b), errors on the OOD test set are higher by at least an order of magnitude, compared to the ID set. A similar pattern is observed for the augmented variant (panel d). While the ID performance of most models is strong, with *DimeNet*$^{++}$ achieving the lowest error, the OOD performance of all models is poor. This generalisation gap is even more pronounced for energy predictions (panels a and c), where *EquiFormerV2* and *DimeNet*$^{++}$ exhibit particularly high OOD errors. A more detailed analysis for this task, including the MAE on forces magnitude, is provided in the appendix.

**Functional group duplication** The results for this task, shown in Figure 4 (panels e and f), reveal a consistent generalisation failure across all architectures. In the case of Forces MAE (panel f), a

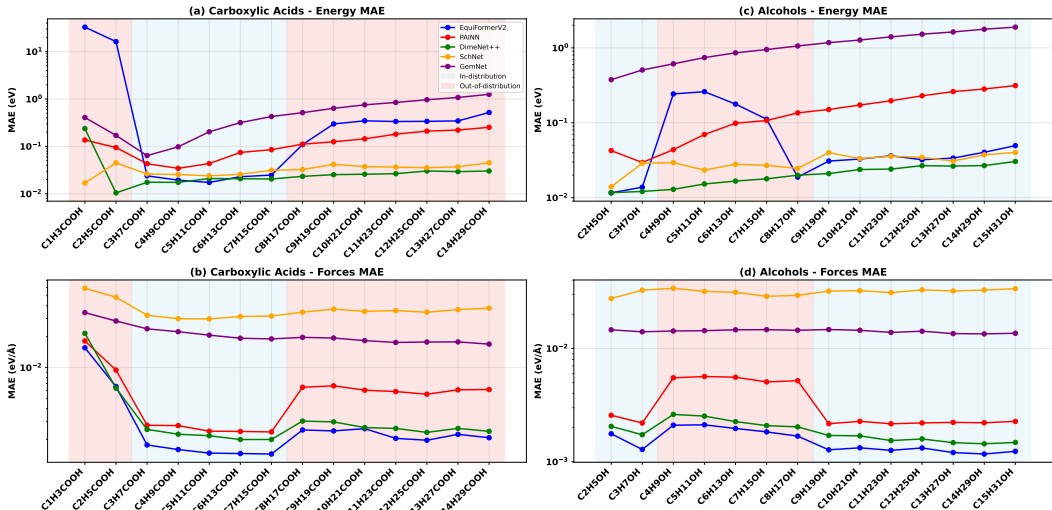

Figure 3: Results for the augmented variant of Length Extrapolation (Task 1). Models were trained on a discontinuous set of *alcohols* with short (2-3 carbons) and long (9-15 carbons) alkyl chains, as well as a set of medium-length *carboxylic acids* (4-8 carbons). The OOD test set evaluates the model's ability to interpolate the properties of medium-chain alcohols and extrapolate to short-chain and long-chain carboxylic acids. The figure displays the *(a)* Energy MAE and *(b)* Forces MAE for carboxylic acids, and the corresponding *(c)* Energy MAE and *(d)* Forces MAE for alcohols. A substantial generalisation error is observed in the OOD regions for both molecular families.

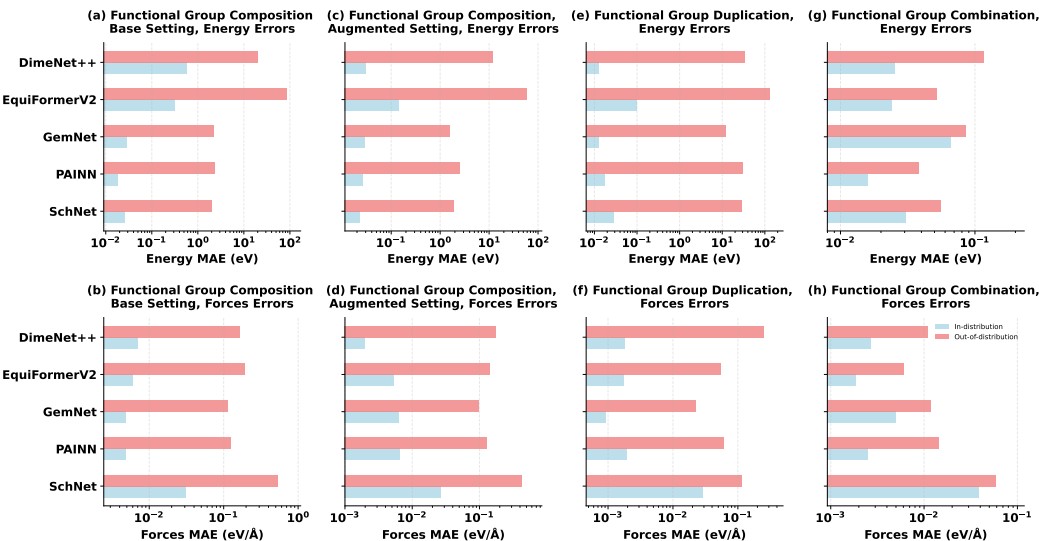

Figure 4: Results for *Functional Group Composition (Task 2)* in its base (a, b) and augmented (c, d) variants, *Functional Group Duplication (Task 3)* (e, f), and *Functional Group Combination (Task 4)* (g, h). Performance is measured by Energy MAE (eV) for the top row (a, c, e, g) and Forces MAE (eV/Å) for the bottom row (b, d, f, h). Each plot compares the ID error (light blue) with the OOD error (red). Across all tasks and models, a substantial generalisation gap is observed.

significant gap between ID and OOD performance is evident for all models. *GemNet* overall performs best, while *SchNet* even struggles on the ID test set. The generalisation gap is even more pronounced for the Energy MAE (panel e), where OOD errors are higher by two orders of magnitude, compared to ID errors. This suggests that the tested architectures do not possess a strong inductive bias for repetition, failing to reuse the learned pattern of a single functional group in a new, duplicated context.

**Functional group combination**   The results, shown in Figure 4 (panels g and h), indicate that while all models still exhibit a clear generalisation gap, this gap is notably smaller than we observed for Functional Group Composition and Functional Group Duplication. For Forces MAE (panel h), *EquiFormerV2* demonstrates the strongest OOD performance. However, for Energy MAE (panel g), *PAINN* performs better, both ID and OOD. This task's intermediate difficulty is particularly informative; the generalisation challenge is significant enough to cause all models to fail, yet not so extreme that their performances become indistinguishable.

## 5   CONCLUSIONS

This work introduces GMD-25, a systematic benchmark designed to evaluate the compositional generalisation capabilities of MLFFs. Through four purposely designed tasks, *length extrapolation*, *functional group composition*, *functional group duplication*, and *functional group combination*, we provide a systematic assessment of how well state-of-the-art MLFFs generalise to molecular configurations outside their training distributions. Our empirical analysis reveals significant limitations in current approaches, with all evaluated models (SchNet, PAINN, DimeNet$^{++}$, GemNet, and EquiFormerV2) showing substantial performance degradation when generalising to out-of-distribution molecular configuration. Errors on out-of-distribution test molecules are often one to two orders of magnitudes higher than on in-distribution examples, indicating fundamental challenges in learning transferable representations of inter-atomic interactions. While substantial generalisation gaps were observed in all four evaluation tasks, this gap was particularly pronounced for *functional group composition* and *functional group duplication*.

The performance of the individual models is highly varied. For instance, EquiFormerV2 performed the best on Length Extrapolation in terms of forces MAE, but it failed completely on energy MAE in the OOD region. SchNet and DimeNet++ performed well on Length Extrapolation in terms of energy MAE. GemNet overall performed best in the OOD region for Functional Group Composition and Functional Group Duplication. For Functional Group Combination, PAINN performed best in terms of energy MAE while EquiFormerV2 performed best in terms of forces MAE.

Our findings highlight a critical gap between the impressive accuracy of MLFFs on standard benchmarks and their ability to extrapolate beyond training distribution. The benchmark serves as a valuable diagnostic tool for identifying architectural biases and guiding the development of more robust, physically-informed models. By focusing on controlled evaluation scenarios rather than simply expanding dataset size or molecular diversity, GMD-25 encourages the development of MLFFs that capture fundamental physical principles rather than dataset-specific patterns. Such models would represent a significant step towards truly predictive force fields capable of accelerating molecular discovery across diverse chemical spaces.

**Reproducibility statement**   To ensure the reproducibility of our results, the complete dataset, including all trajectories and curated data splits, alongside the experimental framework used for our analysis, will be made open-source upon paper acceptance. For those wishing to replicate the data generation process, Section 3.1 details the simulation parameters required to reproduce the trajectories independently, while Section 3.2 describes our custom-built toolkit. An detailed overview of our data splitting policy is available in the appendix. For model training, we utilised the framework mentioned in Section 3.2. The final hyperparameters that were selected for all evaluated models are listed in the appendix.

**Usage of LLMs**   Large Language Models have been used to assist with polishing some of the text in this paper.

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

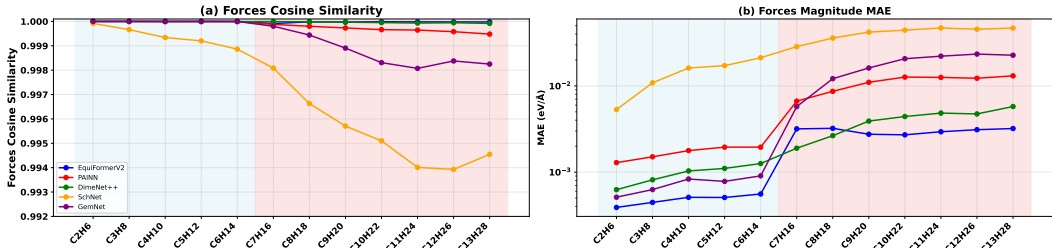

Figure 5: Supplementary force analysis for the base variant of Length Extrapolation (Task 1). Models were trained on linear alkanes with 2-6 carbon atoms (in-distribution, light blue region) and evaluated on unseen chains with 7-13 carbon atoms (out-of-distribution, light red region). This figure decomposes the total force error by showing: (a) the Cosine Similarity, which measures directional accuracy, and (b) the MAE on Force Magnitude, which measures strength accuracy.

# A    APPENDIX

## A.1    ADDITIONAL EVALUATION METRICS

Force MAE conflates errors in both the direction and magnitude of the predicted force vectors. To disentangle these two sources of error, below we present results for two additional metrics: the unitless *Cosine Similarity* and the *MAE on Force Magnitude* (in eV/Å).

**Cosine similarity**   This metric assesses the similarity between the directions of the predicted and true force vectors. A value of 1 indicates perfect alignment, 0 indicates orthogonality, and -1 indicates that the vectors point in opposite directions. The metric is computed by averaging the cosine of the angle between the predicted and true force vectors over all atoms in a given configuration:

$$\text{Cosine Similarity} = \frac{1}{N} \sum_{i=1}^{N} \frac{\hat{\mathbf{F}}_i \cdot \mathbf{F}_i}{||\hat{\mathbf{F}}_i||_2 ||\mathbf{F}_i||_2}$$

where $N$ is the total number of atoms, $\hat{\mathbf{F}}_i$ is the predicted force vector for atom $i$, $\mathbf{F}_i$ is the ground truth force vector, $\cdot$ denotes the dot product, and $|| \cdot ||_2$ is the L2 norm (i.e., the vector magnitude).

**MAE on force magnitude**   This metric evaluates the accuracy of the predicted force strengths (magnitudes) independently of their direction. It quantifies the average absolute error between the magnitude of the predicted force vectors and the magnitude of the true force vectors:

$$\text{MAE}_{\text{mag}} = \frac{1}{N} \sum_{i=1}^{N} \left| ||\hat{\mathbf{F}}_i||_2 - ||\mathbf{F}_i||_2 \right|$$

with $\hat{\mathbf{F}}_i$ and $\mathbf{F}_i$ as before. A lower value for $\text{MAE}_{\text{mag}}$ indicates a more accurate prediction of force strengths.

**Results**   Figures 5–7 summarise the results for the four considered evaluation tasks, in terms of cosine similarity and force magnitude MAE. The results show that errors in the predicted magnitude account for most of the overall errors in the predicted forces. For instance, for Length Extrapolation, models such as EquiFormerV2 and DimeNet++ generalise nearly perfectly in terms of cosine similarity. For Functional Group Combination, we can see nearly perfect OOD generalisation in terms of cosine similarity for all models. For the two most challenging tasks, Functional Group Composition and Functional Group Duplication, we can see that the overall error is a combination of errors in direction and in magnitude. However, even for these tasks, errors in the magnitude prediction play the biggest role.

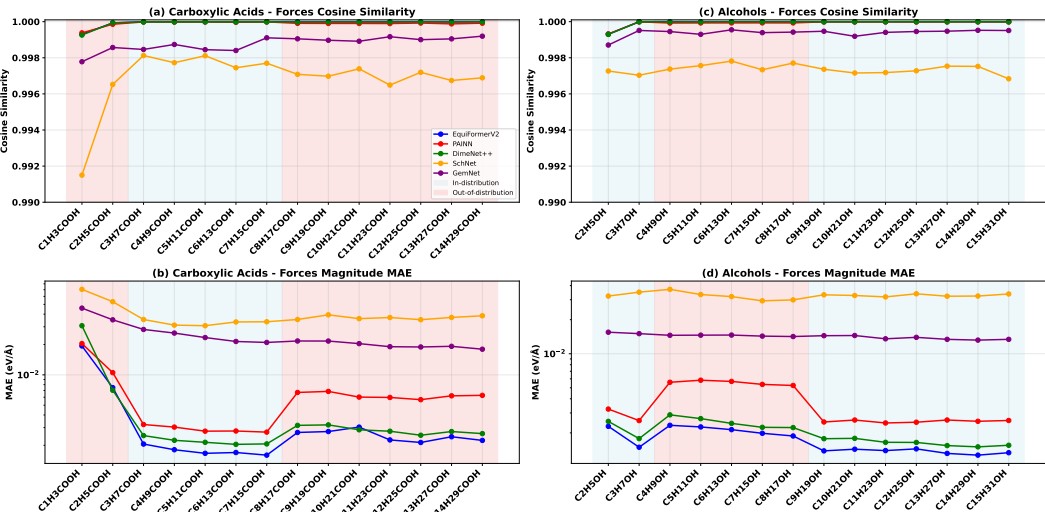

Figure 6: Supplementary force analysis for the augmented variant of Length Extrapolation (Task 1). Models were trained on a discontinuous set of alcohols with short (2-3 carbons) and long (9-15 carbons) alkyl chains, as well as a set of medium-length carboxylic acids (4-8 carbons).

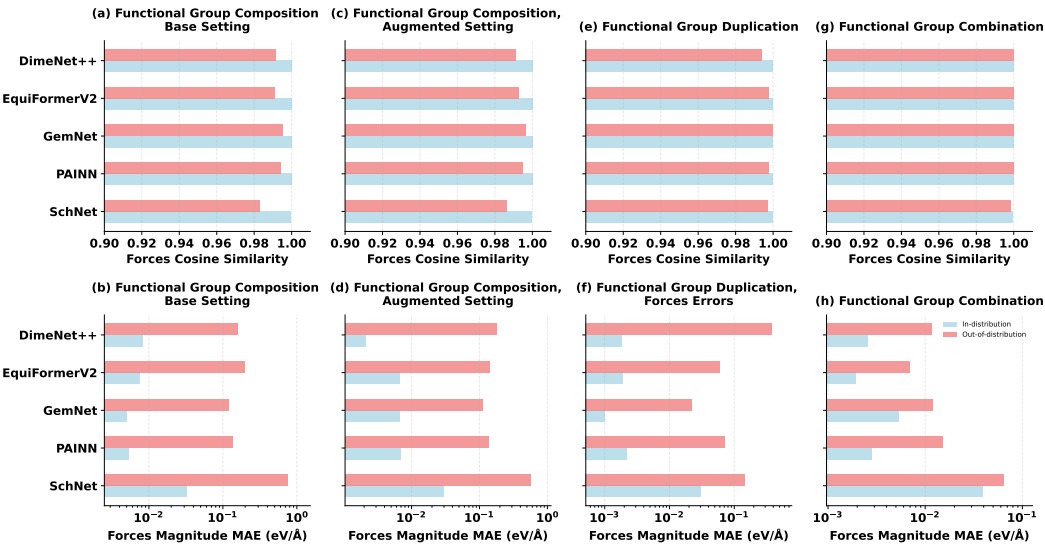

Figure 7: Supplementary force analysis for the Functional Group Composition (Task 2, base and augmented variants), Functional Group Duplication (Task 3) and Functional Group Combination (Task 4), supplementary to Figure 4. The top row (a, c, e, g) displays the Forces Cosine Similarity. The bottom row (b, d, f, h) shows the corresponding Forces Magnitude MAE.

## A.2 THE DATASET

The dataset covers 118 molecules, each with two trajectories. The first trajectory has 2013 snapshots and is primarily used for constructing the training and out-of-distribution test sets. The second trajectory has 500 snapshots, where the first 100 are used for hyperparameter optimisation and the remaining 400 for constructing the in-distribution test sets. The specific molecules used for each task are detailed in Table 1.

While standard protocols for datasets like MD17 often recommend training on a limited number of samples, typically around 1000 snapshots from a single trajectory[4], our approach intentionally deviates from this. The trajectories in our GMD-25 benchmark, unlike those in MD17, which are sampled near equilibrium minima Chmiela et al. (2017), are generated to capture a wide range of off-equilibrium configurations Bigi et al. (2025). Consequently, we utilize 2013 snapshots for the primary trajectory of each molecule. This larger sample size is crucial for two primary reasons. First, it ensures a sufficient representation of the diverse, high-energy states present in our data. Second, by providing a substantial number of examples from each trajectory, it robustly exposes the models to the distinct dynamics of multiple molecular systems.

### A.3 MODEL HYPERPARAMETERS

The hyperparameters for each of the five models evaluated in this study were determined through the two-stage tuning process described in Section 4.2, using the Weights & Biases (W&B) library sweeps for optimisation on Nvidia RTX Ada A6000 GPUs. We used the W&B count parameters to limit the number of optimisation runs in each setting, allocating 30 runs per hyperparameter for each model. This resulted in 180 runs for EquiFormerV2, 120 runs for DimeNet$^{++}$, 150 runs for GemNet-OC, 90 runs for PaiNN, and 120 runs for SchNet. We tuned the following hyperparameters:

- **EquiFormerV2:** sphere channels, FFN hidden channels, max radius, edge channels, number of distance basis, initial learning rate (Table 7)
- **DimeNet$^{++}$:** hidden channels, output embedding channels, cutoff radius, initial learning rate (Table 8)
- **GemNet-OC:** number of radial basis functions, atom embedding size, edge embedding size, cutoff radius, initial learning rate (Table 9)
- **PaiNN:** hidden channels, cutoff radius, initial learning rate (Table 10)
- **SchNet:** hidden channels, number of filters, cutoff radius, initial learning rate (Table 11)

The final optimised parameters used for generating the results are detailed in the tables referenced above, ensuring the reproducibility of our findings.

### A.4 DETAILED RESULTS

This section provides the detailed numerical results that correspond to the figures presented in the paper.

- The exact performance metrics for the base variant of the Length Extrapolation task (Task 1), illustrated in Figure 2 and complemented by the force analysis in Figure 5, are presented in Table 2.
- For the augmented variant of the Length Extrapolation task (Task 1), illustrated in Figure 3 and Figure 6, the numerical results are provided in Table 3 for alcohols and Table 4 for carboxylic acids.
- The aggregated results for the Functional Group Composition (Task 2), Functional Group Duplication (Task 3), and Functional Group Combination (Task 4) tasks, which are visualized in Figure 4 with supplementary force analysis in Figure 7, are provided in Table 5.

---

[4]https://pytorch-geometric.readthedocs.io/en/2.6.0/generated/torch_geometric.datasets.MD17.html

Table 1: Specifications of the dataset splits for each benchmark task, outlining the training and out-of-distribution (OOD) test sets. The in-distribution (ID) test set for each task is drawn from the same molecular distribution as the training set, but consists of data from unseen simulation trajectories.

| Task | Training Set | OOD Test Set |
|---|---|---|
| Task 1: Base | C2-C6 alkanes: Ethane–Hexane | C7-C13 alkanes: Heptane–Tridecane |
| Task 1: Augmented | • C2-C3 & C9-C15 alcohols
• C4-C8 carboxylic acids | • C4-C8 alcohols
• C2-C3 & C9-C15 carboxylic acids |
| Task 2: Base | • C7-C11 complex carbonyls and complex alcohol
• C4-C10 alcohols & aldehydes | C4-C10 carboxylic acids |
| Task 2: Augmented | • C7-C11 complex carbonyls and complex alcohol
• C4-C10 alcohols, aldehydes, amides, and amines | C4-C10 carboxylic acids |
| Task 3 | C5-C10 carboxylic acids | C5-C10 dicarboxylic acids |
| Task 4 | • C2-C9 diamines
• C2-C9 dicarboxylic acids | C2-C9 amino acids |

Table 2: Length extrapolation basic variant model comparison on forces and energy prediction for alkanes. Forces Mean Absolute Error is shown with *F MAE*, Forces Cosine Similarity with *F Cosine* and Forces Magnitude Mean Absolute Error with *F Mag*. Energy is measured in eV and forces in eV/Å. Blue indicates molecules in-distribution and red indicates molecules out-of-distribution for this setting.

| Molecule | Model | F MAE | F Cosine | F Mag | E MAE |
|---|---|---|---|---|---|
| $C_2H_6$ | SchNet | 0.00479 | 0.9999 | 0.00531 | 0.00298 |
| | PAINN | 0.00101 | 1.0000 | 0.00129 | 0.09496 |
| | GemNet | 0.00046 | 1.0000 | 0.00051 | 0.00341 |
| | EquiFormerV2 | 0.00029 | 1.0000 | 0.00039 | 0.00341 |
| | DimeNet$^{++}$ | 0.00057 | 1.0000 | 0.00062 | 0.02874 |
| $C_3H_8$ | SchNet | 0.01036 | 0.9997 | 0.01087 | 0.00710 |
| | PAINN | 0.00124 | 1.0000 | 0.00150 | 0.12240 |
| | GemNet | 0.00058 | 1.0000 | 0.00063 | 0.00772 |
| | EquiFormerV2 | 0.00036 | 1.0000 | 0.00045 | 0.00598 |
| | DimeNet$^{++}$ | 0.00076 | 1.0000 | 0.00081 | 0.05221 |
| $C_4H_{10}$ | SchNet | 0.01539 | 0.9993 | 0.01614 | 0.01461 |
| | PAINN | 0.00149 | 1.0000 | 0.00178 | 0.14741 |
| | GemNet | 0.00076 | 1.0000 | 0.00083 | 0.01242 |
| | EquiFormerV2 | 0.00044 | 1.0000 | 0.00051 | 0.01391 |
| | DimeNet$^{++}$ | 0.00100 | 1.0000 | 0.00103 | 0.06589 |
| $C_5H_{12}$ | SchNet | 0.01712 | 0.9992 | 0.01720 | 0.01751 |
| | PAINN | 0.00164 | 1.0000 | 0.00195 | 0.15998 |
| | GemNet | 0.00071 | 1.0000 | 0.00078 | 0.01377 |
| | EquiFormerV2 | 0.00044 | 1.0000 | 0.00051 | 0.02119 |
| | DimeNet$^{++}$ | 0.00104 | 1.0000 | 0.00110 | 0.07940 |
| $C_6H_{14}$ | SchNet | 0.02119 | 0.9989 | 0.02123 | 0.02302 |
| | PAINN | 0.00166 | 1.0000 | 0.00195 | 0.20522 |
| | GemNet | 0.00085 | 1.0000 | 0.00090 | 0.01799 |
| | EquiFormerV2 | 0.00049 | 1.0000 | 0.00056 | 0.02645 |
| | DimeNet$^{++}$ | 0.00121 | 1.0000 | 0.00126 | 0.09729 |

Table 2 – continued from previous page

| Molecule | Model | F MAE | F Cosine | F Mag | E MAE |
|---|---|---|---|---|---|
| C7H16 | SchNet | 0.02847 | 0.9981 | 0.02857 | 0.02547 |
| | PAINN | 0.00659 | 0.9999 | 0.00666 | 0.34528 |
| | GemNet | 0.00566 | 0.9998 | 0.00576 | 2.43737 |
| | EquiFormerV2 | 0.00221 | 0.9999 | 0.00317 | 3.02265 |
| | DimeNet$^{++}$ | 0.00182 | 1.0000 | 0.00189 | 0.10026 |
| C8H18 | SchNet | 0.03578 | 0.9966 | 0.03592 | 0.02865 |
| | PAINN | 0.00874 | 0.9998 | 0.00866 | 0.56172 |
| | GemNet | 0.01167 | 0.9994 | 0.01214 | 6.04641 |
| | EquiFormerV2 | 0.00239 | 1.0000 | 0.00322 | 7.99460 |
| | DimeNet$^{++}$ | 0.00257 | 1.0000 | 0.00265 | 0.09330 |
| C9H20 | SchNet | 0.04143 | 0.9957 | 0.04214 | 0.02888 |
| | PAINN | 0.01101 | 0.9997 | 0.01102 | 0.87997 |
| | GemNet | 0.01602 | 0.9989 | 0.01616 | 10.41280 |
| | EquiFormerV2 | 0.00232 | 1.0000 | 0.00275 | 12.07372 |
| | DimeNet$^{++}$ | 0.00367 | 1.0000 | 0.00391 | 0.06891 |
| C10H22 | SchNet | 0.04504 | 0.9951 | 0.04436 | 0.03372 |
| | PAINN | 0.01210 | 0.9997 | 0.01267 | 1.20715 |
| | GemNet | 0.02139 | 0.9983 | 0.02069 | 15.42145 |
| | EquiFormerV2 | 0.00233 | 1.0000 | 0.00271 | 16.38867 |
| | DimeNet$^{++}$ | 0.00432 | 1.0000 | 0.00442 | 0.03362 |
| C11H24 | SchNet | 0.04741 | 0.9940 | 0.04707 | 0.03938 |
| | PAINN | 0.01209 | 0.9996 | 0.01256 | 1.55668 |
| | GemNet | 0.02191 | 0.9981 | 0.02215 | 20.95873 |
| | EquiFormerV2 | 0.00254 | 1.0000 | 0.00294 | 22.10180 |
| | DimeNet$^{++}$ | 0.00490 | 0.9999 | 0.00484 | 0.02619 |
| C12H26 | SchNet | 0.04655 | 0.9939 | 0.04553 | 0.03535 |
| | PAINN | 0.01230 | 0.9996 | 0.01228 | 1.76444 |
| | GemNet | 0.02384 | 0.9984 | 0.02342 | 25.92406 |
| | EquiFormerV2 | 0.00266 | 1.0000 | 0.00310 | 27.33273 |
| | DimeNet$^{++}$ | 0.00462 | 0.9999 | 0.00473 | 0.02654 |
| C13H28 | SchNet | 0.04683 | 0.9945 | 0.04693 | 0.04316 |
| | PAINN | 0.01302 | 0.9995 | 0.01309 | 2.21373 |
| | GemNet | 0.02220 | 0.9982 | 0.02273 | 31.43054 |
| | EquiFormerV2 | 0.00274 | 1.0000 | 0.00320 | 31.07756 |
| | DimeNet$^{++}$ | 0.00552 | 0.9999 | 0.00577 | 0.07176 |

Table 3: Length extrapolation augmented variant model comparison on forces and energy prediction for alcohols. Forces Mean Absolute Error is shown with *F MAE*, Forces Cosine Similarity with *F Cosine* and Forces Magnitude Mean Absolute Error with *F Mag*. Energy is measured in eV and forces in eV/Å. Blue indicates molecules in-distribution and red indicates molecules out-of-distribution for this setting.

| Molecule | Model | F MAE | F Cosine | F Mag | E MAE |
|---|---|---|---|---|---|
| C2H5OH | SchNet | 0.02756 | 0.9973 | 0.03234 | 0.01397 |
| | PAINN | 0.00257 | 0.9993 | 0.00325 | 0.04252 |
| | GemNet | 0.01460 | 0.9987 | 0.01550 | 0.37562 |
| | EquiFormerV2 | 0.00177 | 0.9993 | 0.00229 | 0.01159 |

Continued on next page

Table 3 – continued from previous page

| Molecule | Model | F MAE | F Cosine | F Mag | E MAE |
|---|---|---|---|---|---|
| | DimeNet$^{++}$ | 0.00205 | 0.9993 | 0.00253 | 0.01168 |
| C3H7OH | SchNet | 0.03266 | 0.9970 | 0.03500 | 0.02876 |
| | PAINN | 0.00220 | 1.0000 | 0.00257 | 0.02950 |
| | GemNet | 0.01401 | 0.9995 | 0.01505 | 0.50612 |
| | EquiFormerV2 | 0.00128 | 1.0000 | 0.00149 | 0.01380 |
| | DimeNet$^{++}$ | 0.00173 | 1.0000 | 0.00178 | 0.01210 |
| C4H9OH | SchNet | 0.03395 | 0.9974 | 0.03706 | 0.02932 |
| | PAINN | 0.00550 | 0.9999 | 0.00560 | 0.04370 |
| | GemNet | 0.01425 | 0.9995 | 0.01456 | 0.61146 |
| | EquiFormerV2 | 0.00210 | 1.0000 | 0.00233 | 0.24266 |
| | DimeNet$^{++}$ | 0.00262 | 1.0000 | 0.00289 | 0.01292 |
| C5H11OH | SchNet | 0.03185 | 0.9976 | 0.03335 | 0.02336 |
| | PAINN | 0.00565 | 0.9999 | 0.00584 | 0.06966 |
| | GemNet | 0.01434 | 0.9993 | 0.01460 | 0.74061 |
| | EquiFormerV2 | 0.00212 | 1.0000 | 0.00226 | 0.25995 |
| | DimeNet$^{++}$ | 0.00253 | 1.0000 | 0.00267 | 0.01523 |
| C6H13OH | SchNet | 0.03128 | 0.9978 | 0.03198 | 0.02792 |
| | PAINN | 0.00557 | 0.9999 | 0.00570 | 0.09861 |
| | GemNet | 0.01460 | 0.9995 | 0.01463 | 0.85787 |
| | EquiFormerV2 | 0.00197 | 1.0000 | 0.00214 | 0.17750 |
| | DimeNet$^{++}$ | 0.00226 | 1.0000 | 0.00243 | 0.01666 |
| C7H15OH | SchNet | 0.02884 | 0.9973 | 0.02934 | 0.02700 |
| | PAINN | 0.00506 | 0.9999 | 0.00536 | 0.10700 |
| | GemNet | 0.01463 | 0.9994 | 0.01431 | 0.94924 |
| | EquiFormerV2 | 0.00183 | 1.0000 | 0.00198 | 0.11155 |
| | DimeNet$^{++}$ | 0.00208 | 1.0000 | 0.00224 | 0.01785 |
| C8H17OH | SchNet | 0.02940 | 0.9977 | 0.02991 | 0.02462 |
| | PAINN | 0.00519 | 0.9999 | 0.00525 | 0.13553 |
| | GemNet | 0.01446 | 0.9994 | 0.01419 | 1.05923 |
| | EquiFormerV2 | 0.00168 | 1.0000 | 0.00188 | 0.01881 |
| | DimeNet$^{++}$ | 0.00203 | 1.0000 | 0.00223 | 0.01998 |
| C9H19OH | SchNet | 0.03207 | 0.9974 | 0.03317 | 0.03995 |
| | PAINN | 0.00217 | 1.0000 | 0.00250 | 0.15042 |
| | GemNet | 0.01468 | 0.9995 | 0.01444 | 1.17354 |
| | EquiFormerV2 | 0.00127 | 1.0000 | 0.00139 | 0.03081 |
| | DimeNet$^{++}$ | 0.00171 | 1.0000 | 0.00178 | 0.02097 |
| C10H21OH | SchNet | 0.03234 | 0.9972 | 0.03270 | 0.03320 |
| | PAINN | 0.00227 | 1.0000 | 0.00260 | 0.17241 |
| | GemNet | 0.01446 | 0.9992 | 0.01449 | 1.27180 |
| | EquiFormerV2 | 0.00133 | 1.0000 | 0.00144 | 0.03276 |
| | DimeNet$^{++}$ | 0.00169 | 1.0000 | 0.00179 | 0.02377 |
| C11H23OH | SchNet | 0.03109 | 0.9972 | 0.03178 | 0.03577 |
| | PAINN | 0.00216 | 1.0000 | 0.00245 | 0.19655 |
| | GemNet | 0.01385 | 0.9994 | 0.01355 | 1.40246 |
| | EquiFormerV2 | 0.00126 | 1.0000 | 0.00139 | 0.03627 |
| | DimeNet$^{++}$ | 0.00154 | 1.0000 | 0.00165 | 0.02407 |
| C12H25OH | SchNet | 0.03288 | 0.9973 | 0.03385 | 0.03450 |
| | PAINN | 0.00220 | 1.0000 | 0.00248 | 0.22903 |
| | GemNet | 0.01419 | 0.9995 | 0.01396 | 1.52122 |
| | EquiFormerV2 | 0.00132 | 1.0000 | 0.00144 | 0.03210 |

Continued on next page

Table 3 – continued from previous page

| Molecule | Model | F MAE | F Cosine | F Mag | E MAE |
|---|---|---|---|---|---|
| | DimeNet$^{++}$ | 0.00159 | 1.0000 | 0.00165 | 0.02675 |
| C13H27OH | SchNet | 0.03213 | 0.9975 | 0.03223 | 0.03091 |
| | PAINN | 0.00222 | 1.0000 | 0.00260 | 0.26046 |
| | GemNet | 0.01351 | 0.9995 | 0.01340 | 1.63062 |
| | EquiFormerV2 | 0.00121 | 1.0000 | 0.00132 | 0.03375 |
| | DimeNet$^{++}$ | 0.00147 | 1.0000 | 0.00154 | 0.02640 |
| C14H29OH | SchNet | 0.03275 | 0.9975 | 0.03235 | 0.03733 |
| | PAINN | 0.00221 | 1.0000 | 0.00253 | 0.28213 |
| | GemNet | 0.01344 | 0.9995 | 0.01320 | 1.77347 |
| | EquiFormerV2 | 0.00117 | 1.0000 | 0.00127 | 0.04036 |
| | DimeNet$^{++}$ | 0.00144 | 1.0000 | 0.00150 | 0.02681 |
| C15H31OH | SchNet | 0.03366 | 0.9968 | 0.03374 | 0.04015 |
| | PAINN | 0.00227 | 1.0000 | 0.00257 | 0.31360 |
| | GemNet | 0.01361 | 0.9995 | 0.01340 | 1.88979 |
| | EquiFormerV2 | 0.00124 | 1.0000 | 0.00134 | 0.04948 |
| | DimeNet$^{++}$ | 0.00148 | 1.0000 | 0.00155 | 0.03043 |

Table 4: Length extrapolation augmented variant model comparison on forces and energy prediction for carboxylic acids. Forces Mean Absolute Error is shown with *F MAE*, Forces Cosine Similarity with *F Cosine* and Forces Magnitude Mean Absolute Error with *F Mag*. Energy is measured in eV and forces in eV/Å. Blue indicates molecules in-distribution and red indicates molecules out-of-distribution for this setting.

| Molecule | Model | F MAE | F Cosine | F Magnitude | E MAE |
|---|---|---|---|---|---|
| C1H3COOH | SchNet | 0.05903 | 0.9915 | 0.07053 | 0.01691 |
| | PAINN | 0.01815 | 0.9994 | 0.02044 | 0.13633 |
| | GemNet | 0.03424 | 0.9978 | 0.04593 | 0.40684 |
| | EquiFormerV2 | 0.01558 | 0.9993 | 0.01936 | 32.90651 |
| | DimeNet$^{++}$ | 0.02153 | 0.9993 | 0.03073 | 0.23737 |
| C2H5COOH | SchNet | 0.04835 | 0.9965 | 0.05320 | 0.04494 |
| | PAINN | 0.00947 | 0.9999 | 0.01052 | 0.09456 |
| | GemNet | 0.02836 | 0.9986 | 0.03516 | 0.17078 |
| | EquiFormerV2 | 0.00655 | 0.9999 | 0.00747 | 16.28926 |
| | DimeNet$^{++}$ | 0.00632 | 0.9999 | 0.00703 | 0.01044 |
| C3H7COOH | SchNet | 0.03225 | 0.9981 | 0.03541 | 0.02616 |
| | PAINN | 0.00275 | 1.0000 | 0.00322 | 0.04312 |
| | GemNet | 0.02382 | 0.9985 | 0.02822 | 0.06390 |
| | EquiFormerV2 | 0.00177 | 1.0000 | 0.00206 | 0.02389 |
| | DimeNet$^{++}$ | 0.00251 | 1.0000 | 0.00250 | 0.01750 |
| C4H9COOH | SchNet | 0.02988 | 0.9977 | 0.03116 | 0.02547 |
| | PAINN | 0.00273 | 1.0000 | 0.00303 | 0.03440 |
| | GemNet | 0.02231 | 0.9987 | 0.02598 | 0.09800 |
| | EquiFormerV2 | 0.00160 | 1.0000 | 0.00181 | 0.01938 |
| | DimeNet$^{++}$ | 0.00225 | 1.0000 | 0.00224 | 0.01741 |
| C5H11COOH | SchNet | 0.02976 | 0.9981 | 0.03074 | 0.02392 |
| | PAINN | 0.00241 | 1.0000 | 0.00276 | 0.04346 |
| | GemNet | 0.02067 | 0.9984 | 0.02345 | 0.20322 |
| | EquiFormerV2 | 0.00148 | 1.0000 | 0.00167 | 0.01737 |

Table 4 – continued from previous page

| Molecule | Model | F MAE | F Cosine | F Magnitude | E MAE |
|---|---|---|---|---|---|
| | DimeNet$^{++}$ | 0.00218 | 1.0000 | 0.00214 | 0.02095 |
| C6H13COOH | SchNet | 0.03137 | 0.9974 | 0.03349 | 0.02574 |
| | PAINN | 0.00240 | 1.0000 | 0.00277 | 0.07452 |
| | GemNet | 0.01930 | 0.9984 | 0.02144 | 0.31776 |
| | EquiFormerV2 | 0.00146 | 1.0000 | 0.00170 | 0.02290 |
| | DimeNet$^{++}$ | 0.00200 | 1.0000 | 0.00205 | 0.02080 |
| C7H15COOH | SchNet | 0.03169 | 0.9977 | 0.03362 | 0.03134 |
| | PAINN | 0.00237 | 1.0000 | 0.00270 | 0.08525 |
| | GemNet | 0.01899 | 0.9991 | 0.02098 | 0.42688 |
| | EquiFormerV2 | 0.00145 | 1.0000 | 0.00160 | 0.02483 |
| | DimeNet$^{++}$ | 0.00200 | 1.0000 | 0.00207 | 0.02052 |
| C8H17COOH | SchNet | 0.03456 | 0.9971 | 0.03543 | 0.03248 |
| | PAINN | 0.00644 | 0.9999 | 0.00669 | 0.11141 |
| | GemNet | 0.01967 | 0.9991 | 0.02168 | 0.51508 |
| | EquiFormerV2 | 0.00248 | 1.0000 | 0.00268 | 0.10912 |
| | DimeNet$^{++}$ | 0.00302 | 1.0000 | 0.00316 | 0.02332 |
| C9H19COOH | SchNet | 0.03699 | 0.9970 | 0.03939 | 0.04175 |
| | PAINN | 0.00665 | 0.9999 | 0.00685 | 0.12507 |
| | GemNet | 0.01939 | 0.9990 | 0.02165 | 0.63611 |
| | EquiFormerV2 | 0.00242 | 1.0000 | 0.00274 | 0.29776 |
| | DimeNet$^{++}$ | 0.00297 | 1.0000 | 0.00319 | 0.02532 |
| C10H21COOH | SchNet | 0.03520 | 0.9974 | 0.03613 | 0.03744 |
| | PAINN | 0.00603 | 0.9999 | 0.00602 | 0.14429 |
| | GemNet | 0.01829 | 0.9989 | 0.02042 | 0.75389 |
| | EquiFormerV2 | 0.00255 | 1.0000 | 0.00303 | 0.34652 |
| | DimeNet$^{++}$ | 0.00261 | 1.0000 | 0.00285 | 0.02588 |
| C11H23COOH | SchNet | 0.03583 | 0.9965 | 0.03707 | 0.03657 |
| | PAINN | 0.00585 | 0.9999 | 0.00597 | 0.18103 |
| | GemNet | 0.01755 | 0.9992 | 0.01903 | 0.84744 |
| | EquiFormerV2 | 0.00205 | 1.0000 | 0.00226 | 0.33517 |
| | DimeNet$^{++}$ | 0.00256 | 1.0000 | 0.00276 | 0.02651 |
| C12H25COOH | SchNet | 0.03453 | 0.9972 | 0.03526 | 0.03552 |
| | PAINN | 0.00551 | 0.9999 | 0.00567 | 0.21028 |
| | GemNet | 0.01773 | 0.9990 | 0.01891 | 0.96443 |
| | EquiFormerV2 | 0.00197 | 1.0000 | 0.00213 | 0.33812 |
| | DimeNet$^{++}$ | 0.00235 | 1.0000 | 0.00252 | 0.03028 |
| C13H27COOH | SchNet | 0.03664 | 0.9967 | 0.03714 | 0.03703 |
| | PAINN | 0.00608 | 0.9999 | 0.00619 | 0.22115 |
| | GemNet | 0.01777 | 0.9990 | 0.01920 | 1.07803 |
| | EquiFormerV2 | 0.00224 | 1.0000 | 0.00243 | 0.34491 |
| | DimeNet$^{++}$ | 0.00257 | 1.0000 | 0.00274 | 0.02935 |
| C14H29COOH | SchNet | 0.03782 | 0.9969 | 0.03856 | 0.04511 |
| | PAINN | 0.00612 | 0.9999 | 0.00626 | 0.25308 |
| | GemNet | 0.01691 | 0.9992 | 0.01797 | 1.25580 |
| | EquiFormerV2 | 0.00208 | 1.0000 | 0.00224 | 0.51929 |
| | DimeNet$^{++}$ | 0.00240 | 1.0000 | 0.00261 | 0.03019 |

Table 5: Functional group composition, functional group duplication, and functional group combination model comparison on forces and energy prediction for carboxylic acids. Forces Mean Absolute Error is shown with *F MAE*, Forces Cosine Similarity with *F Cosine* and Forces Magnitude Mean Absolute Error with *F Mag*. Energy is measured in eV and forces in eV/Å. ID indicates in-distribution and OOD indicates out-of-distribution.

| Task | Model | F MAE | | Cosine | | F Magnitude | | E MAE | |
|---|---|---|---|---|---|---|---|---|---|
| | | ID | OOD | ID | OOD | ID | OOD | ID | OOD |
| Functional Group Composition Base | SchNet | 0.0311 | 0.5411 | 0.9966 | 0.8306 | 0.0331 | 0.7604 | 0.0259 | 2.0107 |
| | PAINN | 0.0049 | 0.1243 | 0.9999 | 0.9450 | 0.0053 | 0.1381 | 0.0181 | 2.3267 |
| | GemNet | 0.0049 | 0.1151 | 0.9999 | 0.9522 | 0.0050 | 0.1200 | 0.0284 | 2.1902 |
| | EquiFormerV2 | 0.0061 | 0.1909 | 0.9999 | 0.9085 | 0.0076 | 0.1982 | 0.3145 | 85.7026 |
| | DimeNet$^{++}$ | 0.0071 | 0.1639 | 0.9999 | 0.9175 | 0.0082 | 0.1595 | 0.5656 | 20.2845 |
| Functional Group Composition Augmented | SchNet | 0.0271 | 0.4231 | 0.9982 | 0.8630 | 0.0292 | 0.5664 | 0.0227 | 1.914 |
| | PAINN | 0.0066 | 0.1278 | 0.9998 | 0.9504 | 0.0068 | 0.1353 | 0.0268 | 2.485 |
| | GemNet | 0.0064 | 0.0995 | 0.9999 | 0.9633 | 0.0066 | 0.1120 | 0.0292 | 1.603 |
| | EquiFormerV2 | 0.0053 | 0.1427 | 1.0000 | 0.9292 | 0.0066 | 0.1392 | 0.1419 | 59.828 |
| | DimeNet$^{++}$ | 0.0020 | 0.1772 | 1.0000 | 0.9120 | 0.0021 | 0.1797 | 0.0302 | 11.690 |
| Functional Group Duplication | SchNet | 0.02917 | 0.11431 | 0.9979 | 0.9691 | 0.03068 | 0.14575 | 0.02952 | 29.32286 |
| | PAINN | 0.00195 | 0.06001 | 0.9999 | 0.9768 | 0.00217 | 0.07129 | 0.01809 | 30.72007 |
| | GemNet | 0.00092 | 0.02244 | 0.9999 | 0.9978 | 0.00102 | 0.02231 | 0.01289 | 12.27012 |
| | EquiFormerV2 | 0.00176 | 0.05383 | 0.9999 | 0.9746 | 0.00190 | 0.06034 | 0.09839 | 133.7711 |
| | DimeNet$^{++}$ | 0.00180 | 0.24839 | 0.9999 | 0.9406 | 0.00185 | 0.38629 | 0.01280 | 34.86217 |
| Functional Group Combination | SchNet | 0.03841 | 0.05920 | 0.9971 | 0.9875 | 0.03877 | 0.06358 | 0.03033 | 0.05597 |
| | PAINN | 0.00247 | 0.01436 | 0.9999 | 0.9991 | 0.00282 | 0.01507 | 0.01597 | 0.03777 |
| | GemNet | 0.00497 | 0.01165 | 0.9999 | 0.9994 | 0.00535 | 0.01206 | 0.02406 | 0.08581 |
| | EquiFormerV2 | 0.00184 | 0.00598 | 0.9999 | 0.9998 | 0.00193 | 0.00695 | 0.02406 | 0.05212 |
| | DimeNet$^{++}$ | 0.00265 | 0.01085 | 0.9999 | 0.9990 | 0.00257 | 0.01169 | 0.02513 | 0.11633 |

Table 6: Consolidated List of All Molecules in the Dataset by Within Their Group

| Functional Group | Molecules IUPAC Name |
|---|---|
| **Alkanes** | Ethane |
| | Propane |
| | Butane |
| | Pentane |
| | Hexane |
| | Heptane |
| | Octane |
| | Nonane |
| | Decane |
| | Undecane |
| | Dodecane |
| | Tridecane |
| **Alcohols (Primary)** | Ethanol |
| | Propan-1-ol |
| | Butan-1-ol |
| | Pentan-1-ol |
| | Hexan-1-ol |
| | Heptan-1-ol |
| | Octan-1-ol |
| | Nonan-1-ol |
| | Decan-1-ol |
| | Undecan-1-ol |

Table 6 – continued from previous page

| Functional Group | Molecules IUPAC Name |
|---|---|
| | Dodecan-1-ol |
| | Tridecan-1-ol |
| | Tetradecan-1-ol |
| | Pentadecan-1-ol |
| **Aldehydes** | Ethanal |
| | Propanal |
| | Butanal |
| | Pentanal |
| | Hexanal |
| | Heptanal |
| | Octanal |
| | Nonanal |
| | Decanal |
| | Undecanal |
| | Dodecanal |
| | Tridecanal |
| | Tetradecanal |
| | Pentadecanal |
| **Carboxylic Acids** | Ethanoic acid |
| | Propanoic acid |
| | Butanoic acid |
| | Pentanoic acid |
| | Hexanoic acid |
| | Heptanoic acid |
| | Octanoic acid |
| | Nonanoic acid |
| | Decanoic acid |
| | Undecanoic acid |
| | Dodecanoic acid |
| | Tridecanoic acid |
| | Tetradecanoic acid |
| | Pentadecanoic acid |
| **Amines (Primary)** | Ethanamine |
| | Butan-1-amine |
| | Pentan-1-amine |
| | Hexan-1-amine |
| | Heptan-1-amine |
| | Octan-1-amine |
| **Amides (Primary)** | Butanamide |
| | Pentanamide |
| | Hexanamide |
| | Heptanamide |
| | Octanamide |
| **Diamines** | Ethane-1,2-diamine |
| | Propane-1,3-diamine |
| | Butane-1,4-diamine |
| | Pentane-1,5-diamine |
| | Hexane-1,6-diamine |
| | Heptane-1,7-diamine |
| | Octane-1,8-diamine |
| | Nonane-1,9-diamine |
| | Decane-1,10-diamine |
| | Undecane-1,11-diamine |

Table 6 – continued from previous page

| Functional Group | Molecules IUPAC Name |
| --- | --- |
| **Dicarboxylic Acids** | Ethanedioic acid
Propanedioic acid
Butanedioic acid
Pentanedioic acid
Hexanedioic acid
Heptanedioic acid
Octanedioic acid
Nonanedioic acid
Decanedioic acid
Undecanedioic acid
Dodecanedioic acid
Tridecanedioic acid
Tetradecanedioic acid
Pentadecanedioic acid
Hexadecanedioic acid |
| **Amino Acids** | 2-Aminoethanoic acid
3-Aminopropanoic acid
4-Aminobutanoic acid
5-Aminopentanoic acid
6-Aminohexanoic acid
7-Aminoheptanoic acid
8-Aminooctanoic acid
9-Aminononanoic acid
10-Aminodecanoic acid |
| **Complex Multifunctional** | Heptane-1,7-diol
Octane-1,8-diol
Nonane-1,9-diol
Decane-3,9-diol
Decane-4,7-diol
Decane-1,10-diol
Undecane-1,11-diol
Dodecane-1,6,9-triol
Dodecane-1,6,11-triol
Dodecane-1,4,7,10-tetraol
Dodecane-1,4,10,12-tetraol
Undecane-4,9-dione
Dodecanedial
3,4-Dioxodecanal
3,7-Dioxononanedial
3,8-Dioxodecanedial
9-Oxoundecanedial
3,4,7-Trioxononanedial
3,4,8-Trioxodecanedial |

Table 7: EquiformerV2 Model Hyperparameters

| Parameter | Value |
|---|---|
| *Model Architecture* | |
| Number of layers | 8 |
| Sphere channels | 256 |
| Attention hidden channels | 64 |
| Number of heads | 8 |
| Attention value channels | 16 |
| FFN hidden channels | 512 |
| Normalization type | layer_norm_sh |
| *Spherical Harmonics* | |
| $\ell_{\max}$ list | [4] |
| $m_{\max}$ list | [2] |
| Grid resolution | 18 |
| Number of sphere samples | 128 |
| *Graph Structure* | |
| Max neighbors | 20 |
| Max radius | 12.0 Å |
| Max number of elements | 90 |
| Edge channels | 1024 |
| Number of distance basis | 512 |
| *Regularization* | |
| Alpha dropout | 0.1 |
| Drop path rate | 0.1 |
| Projection dropout | 0.0 |
| *Training* | |
| Initial learning rate | 0.0004 |
| Optimizer | AdamW |
| Weight decay | 0.001 |
| Scheduler | LambdaLR (cosine) |
| Gradient clipping | 100 |
| EMA decay | 0.999 |
| Batch size | 8 |
| Max epochs | 100 |

Table 8: DimeNet++ Model Hyperparameters

| Parameter | Value |
|---|---|
| *Model Architecture* | |
| Hidden channels | 512 |
| Output embedding channels | 384 |
| Number of blocks | 3 |
| Number of radial basis functions | 6 |
| Number of spherical harmonics | 7 |
| Number of layers before skip | 1 |
| Number of layers after skip | 2 |
| Number of output layers | 3 |
| *Graph Structure* | |
| Cutoff radius | 6.0 Å |
| Use periodic boundary conditions | False |
| Regress forces | True |
| *Training* | |
| Initial learning rate | 0.0001 |
| Learning rate decay factor | 0.1 |
| Warmup steps | 174,393 |
| Warmup factor | 0.2 |
| Scheduler | LambdaLR (cosine) |
| LR minimum factor | 0.01 |
| Batch size | 4 |
| Max epochs | 100 |

Table 9: GemNet-OC Model Hyperparameters

| Parameter | Value |
|---|---|
| *Model Architecture* | |
| Number of spherical harmonics | 7 |
| Number of radial basis functions | 128 |
| Number of blocks | 6 |
| Atom embedding size | 256 |
| Edge embedding size | 1024 |
| Triplet input embedding size | 64 |
| Triplet output embedding size | 128 |
| Quadruplet input embedding size | 64 |
| Quadruplet output embedding size | 32 |
| Number of layers before skip | 2 |
| Number of layers after skip | 2 |
| Number of atom layers | 3 |
| Number of output layers after atom | 3 |
| *Embedding Sizes* | |
| RBF embedding size | 32 |
| CBF embedding size | 16 |
| SBF embedding size | 64 |
| Atom-interaction input size | 64 |
| Atom-interaction output size | 64 |
| *Graph Structure* | |
| Cutoff radius | 12.0 Å |
| Quadruplet interaction cutoff | 12.0 Å |
| Max neighbors | 30 |
| Max neighbors (quadruplet) | 8 |
| Max neighbors (atom-edge) | 20 |
| Max neighbors (atom) | 1000 |
| *Basis Functions* | |
| RBF type | Gaussian |
| Envelope type | Polynomial (exp=5) |
| CBF type | Spherical harmonics |
| SBF type | Legendre outer |
| *Interactions* | |
| Quadruplet interaction | True |
| Atom-edge interaction | True |
| Edge-atom interaction | True |
| Atom interaction | True |
| Direct forces | True |
| Forces coupled | False |
| *Training* | |
| Initial learning rate | 0.0002 |
| Optimizer | AdamW |
| Weight decay | 0.0 |
| Scheduler | ReduceLROnPlateau |
| LR reduction factor | 0.8 |
| Patience | 3 |
| EMA decay | 0.999 |
| Gradient clipping | 10 |
| Batch size | 8 |
| Max epochs | 100 |

Table 10: PaiNN Model Hyperparameters

| Parameter | Value |
|---|---|
| *Model Architecture* | |
| Hidden channels | 2048 |
| Number of layers | 6 |
| Number of radial basis functions | 64 |
| *Graph Structure* | |
| Cutoff radius | 5.0 Å |
| Max neighbors | 50 |
| Use periodic boundary conditions | False |
| Regress forces | True |
| Direct forces | True |
| *Training* | |
| Initial learning rate | 0.0001 |
| Optimizer | AdamW |
| Weight decay | 0.0 |
| AMSGrad | True |
| Scheduler | ReduceLROnPlateau |
| LR reduction factor | 0.8 |
| Patience | 3 |
| EMA decay | 0.999 |
| Gradient clipping | 10 |
| Batch size | 8 |
| Max epochs | 100 |

Table 11: SchNet Model Hyperparameters

| Parameter | Value |
|---|---|
| *Model Architecture* | |
| Hidden channels | 1024 |
| Number of filters | 256 |
| Number of interactions | 5 |
| Number of Gaussians | 200 |
| *Graph Structure* | |
| Cutoff radius | 6.0 Å |
| Use periodic boundary conditions | False |
| *Training* | |
| Initial learning rate | 0.0005 |
| Optimizer | Adam |
| Learning rate decay factor | 0.1 |
| Warmup steps | 4,687 |
| Warmup factor | 0.2 |
| Scheduler | LambdaLR (cosine) |
| LR minimum factor | 0.01 |
| Batch size | 8 |
| Max epochs | 100 |

