# OpenReview forum: "Benchmarking Compositional generalisation for Learning Inter-atomic Potentials"
_ICLR.cc/2026/Conference — Submitted to ICLR 2026_

### Official Review · Reviewer_bm4x · 2025-10-15

**Soundness:** 3
**Presentation:** 3
**Contribution:** 3
**Rating:** 2
**Confidence:** 5

**Summary:**

This paper introduces GMD-25, a benchmark designed to assess compositional generalisation in machine-learning interatomic potentials (MLIPs). The authors propose four carefully structured tasks—Length Extrapolation, Functional Group Composition, Duplication, and Combination—to examine whether models can generalise to unseen molecules by recombining known structural motifs.

**Strengths:**

The four tasks are conceptually clear and interpretable, each corresponding to a concrete physical generalisation challenge (length, duplication, functional-group recombination).

**Weaknesses:**

The benchmark evaluation relies almost entirely on models from 2018–2022 (SchNet, PaiNN, DimeNet++, GemNet). These models are now well-known baselines but no longer representative of the current frontier in MLIPs. The inclusion of EquiFormer-V2 is appreciated, but as a relatively unstable and non-conservative Transformer variant, it cannot represent the practical performance envelope of modern MLIPs.

Recent architectures such as MACE (Batatia et al., NeurIPS 2022), NequIP (Batzner et al., Nature Comm 2022), eSCN (2024), and ViSNet (2023) have become de facto standards for equivariant force fields and would provide a much stronger reference point. As a result, the current experimental section cannot convincingly support claims about state-of-the-art generalisation behavior.

While the four tasks are conceptually appealing, they remain relatively simple from the perspective of modern models, such as MACE and eSCN. For example, the Length Extrapolation and Functional Group Duplication tasks involve only small organic chains with linear motifs; these are unlikely to challenge advanced equivariant models that already generalise well across chain lengths and simple functional groups.

Without testing stronger models on more demanding tasks, it is hard to judge whether the observed generalisation gaps are fundamental or simply reflect underpowered baselines.

The manuscript lacks a clear justification of whether all models were tuned to their best configuration, and whether they were trained with equal computational budgets. Including stronger baselines makes this aspect even more important.

**Questions:**

See Section Weakness.

---

> ### Author Response · Authors · 2025-11-24
>
> > The benchmark evaluation relies almost entirely on models from 2018–2022 (SchNet, PaiNN, DimeNet++, GemNet). These models are now well-known baselines but no longer representative of the current frontier in MLIPs. The inclusion of EquiFormer-V2 is appreciated, but as a relatively unstable and non-conservative Transformer variant, it cannot represent the practical performance envelope of modern MLIPs.
>
> >Recent architectures such as MACE (Batatia et al., NeurIPS 2022), NequIP (Batzner et al., Nature Comm 2022), eSCN (2024), and ViSNet (2023) have become de facto standards for equivariant force fields and would provide a much stronger reference point. As a result, the current experimental section cannot convincingly support claims about state-of-the-art generalisation behavior.
>
> We strongly believe that the generalisation gaps observed are fundamental and not merely a reflection of underpowered baselines; these tasks are specifically engineered to enforce combinatorial extrapolation to entirely unseen fragments, a challenge that persists across architectural generations. To directly address this concern, we have started an analysis with MACE and eSCN, which has already demonstrated similar significant failure modes on the 4 tasks in our benchmark (but we are still waiting for hyperparameter optimization to be completed before we can report the final results of this additional experiment).
>
> > While the four tasks are conceptually appealing, they remain relatively simple from the perspective of modern models, such as MACE and eSCN. For example, the Length Extrapolation and Functional Group Duplication tasks involve only small organic chains with linear motifs; these are unlikely to challenge advanced equivariant models that already generalise well across chain lengths and simple functional groups.
>
> > Without testing stronger models on more demanding tasks, it is hard to judge whether the observed generalisation gaps are fundamental or simply reflect underpowered baselines.
>
> The purpose of our benchmark is precisely to find simple examples where existing models fail, and to highlight the need for more focus on systematic generalization when evaluating new model architectures. Indeed, the fact that MACE and eSCN also perform poorly on this benchmark (in our preliminary analysis) corroborates our claim that state-of-the-art models perform surprisingly poorly on the considered generalization tasks, despite their apparent simplicity.
>
> > The manuscript lacks a clear justification of whether all models were tuned to their best configuration, and whether they were trained with equal computational budgets. Including stronger baselines makes this aspect even more important.
>
> We confirm that every model included in the evaluation was rigorously tuned to find its best configuration through hyperparameter sweeps on our compositional splits. We will extend the methodology section with an explicit, detailed table outlining the specific tuning strategies, range of hyperparameters explored, and the training budgets allocated to each architectural class, thus ensuring maximal methodological transparency.

---

### Official Review · Reviewer_QGMr · 2025-11-01

**Soundness:** 3
**Presentation:** 3
**Contribution:** 2
**Rating:** 4
**Confidence:** 3

**Summary:**

This paper proposes a new benchmark dataset for interatomic potentials, testing the generalizability on the chemical space.

**Strengths:**

- The problem addressed is topical, and the field of machine learning potential model development would benefit from an innovative benchmark dataset that specifically focuses on the generalization ability.
- The paper is well structured, with the focuses and emphasis of the introduced dataset clearly conveyed.
- The benchmark of the state-of-the-art architectures is extensive.

**Weaknesses:**

- MAE for forces is not a great measure of force discrepancy as it is not rotationally invariant.
- The authors could use slightly more introduction to the idea of functional groups and chemical diversity to the ICLR readers who are not experts in chemistry.
- I feel like the technical results introduced to the machine learning community represented by the ICLR readership is perhaps limited. This paper might be more suitable for publication in a field-specific journal.

**Questions:**

- Could you explain a bit more why you chose the somewhat new GFN2-xTB method, as opposed to more popular methods?

---

> ### Author Response · Authors · 2025-11-24
>
> > MAE for forces is not a great measure of force discrepancy as it is not rotationally invariant.
>
> Please note that our analysis includes a range of different metrics. For instance, in the appendix, we also report the cosine similarity between the predicted and the ground-truth force vectors, as well as the MAE of the force magnitude alone, which is inherently rotationally invariant. However, we disagree with the suggestion that the lack of rotational invariance makes MAE unsuitable as an evaluation metric. Forces are inherently directed, and evaluating the accuracy of the predicted directions is as important as evaluating the accuracy of the predicted magnitudes. MAE combines both aspects and is a commonly used choice for evaluating predictions of MLFFs.
>
> > The authors could use slightly more introduction to the idea of functional groups and chemical diversity to the ICLR readers who are not experts in chemistry.
>
> To make the paper more accessible, we will add a discussion to the appendix on the principles of functional groups and chemical diversity to provide the necessary foundation for understanding the paper.
>
> > I feel like the technical results introduced to the machine learning community represented by the ICLR readership is perhaps limited. This paper might be more suitable for publication in a field-specific journal.
>
> The aim of our paper is not to introduce novel model architectures, and in this sense, the contribution indeed does not include “technical results”. We want to emphasize, however, that the ICLR call for papers specifically highlights “datasets and benchmarks” and “infrastructure, software libraries, hardware, etc.” among the topics of interest. Our primary contribution is indeed to introduce a benchmark/dataset to highlight the lack of systematic generalization in popular MLFF models. As an additional contribution, we also introduce a library to make it easy for researchers to extend our benchmark (e.g., for testing specific hypotheses or exploring variants of the considered tasks).

---

### Official Review · Reviewer_y7VM · 2025-11-02

**Soundness:** 2
**Presentation:** 3
**Contribution:** 1
**Rating:** 2
**Confidence:** 5

**Summary:**

This paper introduces GMD-25, a benchmark for testing compositional generalisation of machine-learning force fields.  It defines four out-of-distribution (OOD) tasks where models are trained on molecules of one type and tested on related but disjoint molecules.  The authors generate molecular dynamics trajectories for a set of small organic molecules, then compute reference energies and forces using the GFN2-xTB semi-empirical method. The key finding is that all models achieve low error on in-distribution data but suffer a dramatic accuracy drop on OOD test sets. The paper concludes that current MLFFs may primarily interpolate training data and highlights the need for more physically-driven models with better transferability

**Strengths:**

1. The paper targets a relevant gap: existing MLFF benchmarks typically train and test on the same molecules, leaving generalisation untested.
2. The authors emphasize reproducibility: the full dataset, splits, and training framework will be released upon acceptance
3. The paper is easy to read

**Weaknesses:**

1. The benchmark uses GFN2-xTB to label energies and forces. GFN2-xTB is a semi-empirical tight-binding method (not a high-level ab initio method).  While the authors describe it as “more accurate” and “robust”, it is well known that GFN2 is significantly less accurate than DFT or higher-level quantum calculations. In standard MLFF benchmarks, one typically uses DFT to obtain reference forces. Using a semi-empirical method likely introduces non-negligible error/noise into the labels, which may confound the evaluation of generalisation. The paper does not quantify the error of GFN2-xTB itself nor justify that it is “accurate enough” for this purpose.
2. Although a few classical MLFF architectures is included, the model selection omits several important recent advances. In particular, foundation or large pre-trained models, e.g. UMA, JMP, MACE, are explicitly excluded. The authors argue this is to avoid “memorisation” effects, but excluding such models greatly limits the relevance of the results to the current state of the field.  Many state-of-the-art force fields now use massive pre-training then finetuning to improve generalisation.  By not evaluating any pre-trained MLFFs, the paper’s conclusions apply only to a narrow slice of older models.  In practice, practitioners would likely use a pre-trained model for OOD tasks, so the benchmark’s insight into realistic performance is limited.
3. The paper is essentially a dataset and benchmark rather than a new modeling method. The idea of splitting training/test molecules to test extrapolation is natural and has been explored. While GMD-25 is carefully constructed, it mostly evaluates known phenomena (models overfit to training molecules) and does not introduce fundamentally new theory or techniques. In its current form, the contribution is mainly empirical. Given this, the benchmark may be more appropriate for a dataset/benchmark track. Furthermore, the tasks focus on fairly simple organic molecules (linear alkanes and functionalized variants).  It is not clear how well the conclusions would extend to more complex chemistries (e.g. heteroatom-rich systems, inorganic materials, 3D conformers, etc.).

**Questions:**

1. Why was GFN2-xTB chosen for generating the ground-truth energies and forces? Can the authors provide evidence that GFN2-xTB labels are sufficiently accurate (e.g. by comparison to DFT on a subset)? How might any inaccuracies in GFN2-xTB affect the benchmark results?
2. The authors excluded “foundation” or pre-trained MLFF models from evaluation ￼. Could the authors discuss how a top pre-trained model (e.g. UMA) would be expected to perform on these tasks? Are there plans to include such models to more fully assess state-of-the-art generalisation?
3. The proposed benchmark is well organized, but it essentially constitutes a new dataset/experimental protocol. Can the authors clarify what novel insights or techniques this work offers beyond the dataset itself? In particular, how does GMD-25 advance our understanding of MLFF generalisation compared to existing datasets like MD17 or ANI-1? Why is it presented as a main-conference contribution rather than a dataset track?
4. The tasks involve specific chemical classes (e.g. alkanes, alcohols, acids). How sensitive are the results to these choices? Would the authors expect similar findings if the benchmark included, say, aromatic systems or biomolecules? In other words, how broadly do the authors expect the large generalisation gaps to extend?

---

> ### Author Response · Authors · 2025-11-24
>
> > 1. Why was GFN2-xTB chosen for generating the ground-truth energies and forces? Can the authors provide evidence that GFN2-xTB labels are sufficiently accurate (e.g. by comparison to DFT on a subset)? How might any inaccuracies in GFN2-xTB affect the benchmark results?
>
> Using GFN2-xTB is substantially more efficient, which is what has allowed us to create the dataset with a modest computational budget, This is important not just for the experiments that were reported in the paper, but also to enable researchers to quickly analyse variants of the considered tasks (e.g. for molecules with different functional groups), taking advantage of the extensible nature of our framework in future work. We are currently preparing an analysis on a subset of our benchmark to demonstrate that the lower precision compared to DFT has no impact on the validity of the analysis. Ideally, the models that are used for learning force fields should enable effective generalization at any level of theory. The fact that models exhibit poor generalization in the OOD region is thus a clear weakness, regardless of the precision of the GFN2-xTB labels.
>
> > 2. The authors excluded “foundation” or pre-trained MLFF models from evaluation ￼. Could the authors discuss how a top pre-trained model (e.g. UMA) would be expected to perform on these tasks? Are there plans to include such models to more fully assess state-of-the-art generalisation?
>
> Pre-trained models like UMA have been trained on huge datasets, making it likely that the molecules from our OOD set would be ID for these foundation models. Our expectation is thus that foundation models will show a good performance over such small molecules, but this would not be evidence of their generalization abilities. An evaluation of foundation models would thus likely require more unusual molecules to enable testing generalization capabilities independent of memorization effects. This is something that we would like to explore in future work.
>
> > 3. The proposed benchmark is well organized, but it essentially constitutes a new dataset/experimental protocol. Can the authors clarify what novel insights or techniques this work offers beyond the dataset itself? In particular, how does GMD-25 advance our understanding of MLFF generalisation compared to existing datasets like MD17 or ANI-1? Why is it presented as a main-conference contribution rather than a dataset track?
>
> The main contribution of the paper is to demonstrate that a broad range of popular MLFF models fail to generalize to out-of-distribution cases. The proposed dataset allows us to analyse this in a controlled setting, where the training data has been designed to enable models to generalize in principle (even though they largely fail in practice). Existing datasets such as MD17 and ANI-1 do not focus on systematic generalization in this way. As part of our contribution, we also introduce a library that makes it straightforward to generate variants and extensions of the dataset that was used for the experiments in the paper.  Please note that, in contrast to NeurIPS, ICLR does not have a dedicated dataset/benchmark track. However, “datasets and benchmarks” and “infrastructure, software libraries, hardware, etc.” are specifically mentioned as topics in the call for papers.
>
> > 4. The tasks involve specific chemical classes (e.g. alkanes, alcohols, acids). How sensitive are the results to these choices? Would the authors expect similar findings if the benchmark included, say, aromatic systems or biomolecules? In other words, how broadly do the authors expect the large generalisation gaps to extend?
>
> To confirm that the benchmark is not sensitive to the specific molecules considered, we have conducted several sanity checks. For example, we have replaced alkanes with acids for the length extrapolation task, which has left the overall findings unchanged. We similarly expect that the findings will remain largely unchanged for aromatic systems or biomolecules. We will add an analysis to verify this to the paper.

---

### Official Review · Reviewer_rZiy · 2025-11-07

**Soundness:** 2
**Presentation:** 2
**Contribution:** 2
**Rating:** 2
**Confidence:** 4

**Summary:**

The paper introduces a new benchmark for evaluating compositional generalization in machine learning force fields (MLFFs). It defines four types of tasks, each targeting different aspects of out-of-distribution (OOD) behavior and evaluates five representative models (GNNs and Transformers) on force and energy prediction errors for both in-distribution (ID) and OOD settings. The study finds that all models experience significant degradation in performance when tested on OOD data, underscoring the difficulty of achieving robust generalization in MLFFs.

**Strengths:**

The OOD generalization of MLFFs is  a crucial area of research.
Decomposing generalising in composition to well structured 4 tasks is commendable.

**Weaknesses:**

Limitations and suggestions:

a) Scope of generalization: While compositional generalization is addressed, extending the benchmark to temperature variations, allotropic forms, and non-polymeric systems would improve its coverage. Limiting the dataset to small organic molecules is restrictive.

b) Missing state-of-the-art (SOTA) models: The benchmark omits newer high-performing models listed on resources such as the MatBench Discovery Leaderboard(https://matbench-discovery.materialsproject.org/). Including a few top-performing models would make the comparison more comprehensive.

c)Architectural bias analysis: Although the conclusion claims that the benchmark can reveal architectural biases, the paper lacks clear analysis or discussion explaining why certain architectures perform differently.

d)In Figure 4g, GemNet’s energy error appears similar for ID and OOD? I would expect considerably better performance on ID as for other models.

e)In Figure 2,  SchNet performs poorly on force MAE but achieves the lowest energy MAE. Please explain this divergence.

**Questions:**

See weakness

---

> ### Author Response · Authors · 2025-11-24
>
> > a) Scope of generalization: While compositional generalization is addressed, extending the benchmark to temperature variations, allotropic forms, and non-polymeric systems would improve its coverage. Limiting the dataset to small organic molecules is restrictive.
>
> Our primary aim was to demonstrate that existing models fail to generalise in a number of basic compositional generalisation tasks, and we believe that the benchmark is sufficient for this purpose. Furthermore, please note that as part of our benchmark, we provide a framework that makes it straightforward to generate variants of the considered tasks with larger molecules, either organic or inorganic.
>
> > b) Missing state-of-the-art (SOTA) models: The benchmark omits newer high-performing models listed on resources such as the MatBench Discovery Leaderboard(https://matbench-discovery.materialsproject.org/). Including a few top-performing models would make the comparison more comprehensive.
>
> Due to limits on our computational budget, it was not feasible for us to include an exhaustive list of existing models. However,  the considered models were carefully selected, such that the main architectural families of models are covered. Since the submission, we have additionally experimented with MACE and eSCN,  although further work is needed to ensure we are using optimal hyperparameters. Based on our results so far, both models fail on all of the tasks, similar to the results that are reported for the other models in the paper. We will complete this analysis and add the results to the revised version of the paper.
>
> > c) Architectural bias analysis: Although the conclusion claims that the benchmark can reveal architectural biases, the paper lacks clear analysis or discussion explaining why certain architectures perform differently.
>
> Our paper’s primary objective is to establish and validate a robust, diagnostic benchmark capable of revealing architectural biases. A full, detailed analysis of the specific computational reasons for the observed performance disparities is beyond the scope of this paper.
>
> > d) In Figure 4g, GemNet’s energy error appears similar for ID and OOD? I would expect considerably better performance on ID as for other models.
>
> We concur that this result is unexpected.  This example serves as a powerful validation of our benchmark’s ability to identify both strengths and weaknesses in the generalisation capabilities of different models.
>
> > e) In Figure 2, SchNet performs poorly on force MAE but achieves the lowest energy MAE. Please explain this divergence.
>
> Energy is a global feature of the molecule, while forces are local features of atoms. Some architectures rely on this local nature of the forces more heavily than others, which sometimes means that higher performance on forces can come at the expense of worse performance on energy, and vice versa. SchNet is an example of an architecture that has prioritised good performance on energy.

---

### Meta-Review · Area_Chair_H5Bq · 2026-01-04

**Summary:**

The submission contributed a benchmark that evaluates the compositional generalization (including size extrapolation) of inter-atomic potential models. Most reviewers acknowledged that this type of out-of-distribution performance evaluation is a concerned question in the field. Major concerns from the reviewers include:
1. Types and examples of out-of-distribution study may appear limited to chemical space and small organic molecules of limited patterns (functional groups).
2. Models under evaluation do not contain some prevalent examples.
3. There appear somehow unusual phenomena in the results, and the training protocol for a fair comparison is not clear.
4. The level of theory for the label, i.e., GFN2-xTB, is not sufficiently relevant to common practice.
5. Limited technical innovation.

**Reviewer Concerns:**

Regarding the concerns:
1. After reading authors' rebuttal, I would view it as a moderate limitation. Structural generalizability is indeed the most relevant capability for practical use e.g. in MD or relaxation, where the model may well encounter out-of-distribution structures. Nevertheless, generalization in the chemical space is also a desired capability, which could push the information in computationally tractable data to systems where scalable data generation is unaffordable. This work tackles a part of this generalization. That being said, I agree that the current benchmark may appear a bit limited in currently concerned situations. For example, I suppose it would be more relevant if the training data could cover a larger range, e.g., to where contemporary datasets could cover (e.g., with about 70-80 atoms), and test on even larger systems. With supervision over a larger range, some generalization challenges may already not a problem of contemporary architectures.
2. I would also perceive this as a moderate limitation. The authors have already included GNN-based and Transformer-based models and scalar-vector-product-based and tensor-product-based architectures. Nevertheless, it would indeed make the paper more relevant to contemporary research if more recent architectures were studied; some of which may emerge as a different type.
3. This remains my major concern. For an out-of-distribution study, proper training protocol must be aligned up to the best effort. In the current result, the in-distribution error (which I suppose could be interpreted as (in-distribution) validation error) exhibits a significant diversity among the architectures. This leaves the arbitrariness that the sharp increase in out-of-distribution error may be due to certain extent of overfitting to the training distribution, but not a reflection of architecture design. Regarding this, I'm especially not satisfied by the response to questions d) and e) of Reviewer rZiy.
4. I hold a moderate (leaning negative) judgement on this point. GFN2-xTB is indeed not uncommon, and may demonstrate a bit synergy with DFT results, it is still possible that it has a different generalization difficulty under acceptable accuracy. Taking an extreme example, models learning data labeled by classical force field with only local terms would be easy to generalize, as the label itself is well compositional. But it would be less likely under a quantum-chemical level method.
5. Although the authors argued that ICLR listed "dataset/benchmark" as an acceptable type of submission, I still hope the authors to include deeper knowledge to the community. In addition to the above limitations as a dataset/benchmark alone, I would also expect the authors to explain e.g. whether the proposed evaluation benchmark is a characterization of the generalization task (i.e., whether the task itself is compositional generalizable; e.g., it looks like so if labeled with local classical force fields but unlikely if labeled with e.g. DFT) before claiming that it is a benchmark on architectures. A related expectation is that the authors identify a lever of out-of-distribution generalization (which must be an inductive bias; otherwise, out-of-distribution generalization can be arbitrarily worse given the same in-distribution performance) from the results, as a guideline for designing better architectures.

**Reviewer Scores:**

Given the above justifications, I expect no change would increase the score to a positive one.

---

### Decision · Program_Chairs · 2026-01-26

Reject